

# Original Research Article

# Developing a Decision Support Tool for Assessing Land Use

# Change and BMPs in Large Ungauged Watersheds

Junyu Qi[a], Sheng Li[a,b], Charles P.-A. Bourque[a], Zisheng Xing[a,b], and Fan-Rui Meng[a,*]

[a] Faculty of Forestry and Environmental Management, University of New Brunswick,

P.O. Box 44400, 28 Dineen Drive, Fredericton, NB, E3B 5A3,

Canada

[b]Potato Research Centre, Agriculture and Agri-Food Canada, P.O. Box 20280, 850

Lincoln Road, Fredericton, NB, E3B 4Z7, Canada

*Corresponding author: Fan-Rui Meng, Tel.: +1 506 453 4921, E-mail: fmeng@unb.ca



**Abstract**
A simple decision support tool (DST) was developed to evaluate impacts of land use
change and best management practices (BMPs) on water resources for large ungauged
watersheds in New Brunswick, Canada. It was developed based on statistical equations
derived from Soil and Water Assessment Tool (SWAT) simulations applied to a small
experimental watershed in northwest New Brunswick. The DST was subsequently tested
against field measurements and SWAT-model simulations for a larger watershed. Results
from DST reproduced both field data and model simulations of annual stream discharge
and sediment and nutrient loadings fairly well. The relative error of mean annual
discharge and sediment and nutrient loading were within -52 to +27%. Compared with
SWAT, DST has fewer input requirements and can be applied to multiple watersheds
without additional calibration. Also, scenario analyses with DST can be directly
conducted for different combinations of land use and BMPs without complex model
setup procedures.
**Keywords**: multiple regression; hydrological model; erosion; nitrate leaching;
geographic information system





## 1. Introduction

Pollution from nonpoint sources poses a significant threat to ecosystems and plant and animal communities (Vörösmarty et al., 2010). Nonpoint sources of sediment, nutrients, and pesticides, primarily from agricultural lands, have been identified as major contributors to water quality degradation (Ongley et al., 2010; Zhang et al., 2004). These pollutants are difficult to control because they come from many sources (Quan and Yan, 2001). Practices such as strip cropping, terracing, crop rotation, and nutrient management can be developed to prevent soil erosion and reduce the movement of nutrients and pesticides from agricultural lands to aquatic ecosystems (D'Arcy and Frost, 2001). These pollution-prevention methods, known as best management practices (BMPs), are intended to minimize the negative environmental impact of agricultural activities, while maintaining land productivity. Reliable information on the impacts of land use change and BMPs on water quantity and quality is critical to watershed management (Panagopoulos et al., 2011).

Many studies have been conducted to evaluate the impact of land use change and BMPs on water quality based on field experiments (Novara et al., 2011; Pimentel and Krummel, 1987; Sadeghi et al., 2012; Turkelboom et al., 1997; Urbonas, 1994). Monitoring systems have been established to assess the impact of land use change and BMPs on water resources in order to capture the spatial and temporal variation in soil, climate, and topographic conditions in watersheds (Veldkamp and Lambin, 2001). Statistical models developed from field data from small watersheds are usually assumed to apply to large watersheds (Bloschl and Grayson, 2001; Blöschl and Sivapalan, 1995). Although it is not difficult to quantify soil erosion and chemical loadings in experimental



plots, it is time-consuming and expensive (Mostaghimi et al., 1997). Clearly, it is not
practical to conduct field experiments for every possible combination of land use and
BMPs, under different biophysical conditions. As a result, it is unlikely sufficient field
data could be obtained to develop management plans and conduct cost-benefit analyses.
In addition, statistical models could be potentially derived from experiments; however, it
is difficult to establish cause-and-effect relationships between BMPs and water quality
variables under varied biophysical conditions or to quantify the impact of combined land
use and BMPs on water quality at the watershed scale (Renschler and Lee, 2005).

Process-based models of hydrology can be used to extrapolate field data to fill data

gaps (Borah and Bera, 2003; Borah and Bera, 2004; Singh, 1995; Singh and Frevert,
2005; Singh and Woolhiser, 2002).  These process-based models provide quantitative
information that is usually difficult to obtain from field experiments (Borah et al., 2002).
For example, ANSWERS (Beasley et al., 1980), CREAMS (Knisel, 1980), GLEAMS
(Leonard et al., 1987), AGNPS (Young et al., 1989), EPIC (Sharpley and Williams,
1990), and SWAT (Arnold et al., 1998) have been used to understand surface runoff, soil
erosion, nutrient leaching, and pollutant-transport processes. However, these process-
based models require extensive input data and complex calibration procedures (Liu et al.,
2015); watersheds with sufficient data to calibrate and validate these models are normally
small, resulting in lack of representation at large spatial scales. Furthermore, once a
model is calibrated, parameters become watershed-specific, which cannot be easily
extended to other watersheds. In addition, these models require specialized expertise,
which prevents non-expert decision makers and economists to use them (Viavattene et al.,

2008).





A decision support tool could be developed by combining "decision rules" with
geographic information systems (GIS) for water quality assessment in large ungauged
watersheds. The "decision rules" could be based on regression equations derived from
field experiments (Renschler and Harbor, 2002), or they could be defined simply as
constants based on expert knowledge. Alternatively, simulations from a well-calibrated
hydrological model could be used to develop statistical equation-based "decision rules".
Apart from defining "decision rules" at each grid cell, to assess water quantity and
quality in streams or at subbasin/watershed outlets, the decision support tool should
consider discharge, sediment, and nutrient routing within the watershed. For example, a
commonly used routing mothed for sediments is the sediment-delivery ratio (SDR)
method, which is widely employed in many GIS-based erosion models (May and Place,
2010; Wilson et al., 2001; Zhao et al., 2010). For discharge, a simple summation routing
at the outlet produces acceptable accuracy for small- and medium-sized watersheds,
considering that there is negligible water losses from surface runoff and stream flow. For
large watersheds, water losses are generally greater. These water losses can be estimated
using simple linear equations. The annual export of nutrients from watersheds (via the
nutrient-delivery ratio) has been studied empirically in many studies as nutrient loading
per land area (Beaulac and Reckhow, 1982; Endreny and Wood, 2003; Reckhow and
Simpson, 1980).
A decision support tool developed based on "decision rules" is generally flexible and
easy for decision makers and economists to use (Endreny and Wood, 2003). However,
their practicality in normal circumstances, particularly with respect to their level of
accuracy, needs to be evaluated. In addition, in order to provide sufficient "decision rules"



with reasonable accuracy, fully validated hydrological models are required to be able to
fill data gaps in field experiments. The present study used the Soil and Water Assessment
Tool (SWAT) to provide modelled data in the development of the decision support tool.
The main objective of the present study is to develop a simple decision support tool with
the intent to evaluate the impact of land use change and BMPs on water resources in a
large ungauged watershed in New Brunswick, Canada. This paper presents the
development and testing of a decision support tool using data from two watersheds in the
potato-belt of New Brunswick; one small experimental watershed, with extensive
monitoring and field survey data, and a larger watershed containing the smaller
watershed.
**2.   Materials and Methods**
The general framework of the study is illustrated in Fig. 1. Specifically, this involves:
(1) setting up, calibrating, and validating SWAT for a small experimental watershed; (2)
developing statistical equations based on SWAT-model simulations for different
combinations of land use and BMPs; (3) integrating the statistical equations into a
decision support tool with the aid of ArcGIS; and (4) testing the decision support tool
against field measurements and model simulations of water quantity and quality for a
large watershed.






Fig. 1 Information flow in development of the decision support tool.



## 2.1 Study Sites and Data Collection


The large watershed of this study is the Little River Watershed (LRW), located in the
Upper Saint John River Valley of northwestern New Brunswick, Canada (Fig. 2). It
covers an area approximately 380 km$^2$ with a mixture of agricultural (16.2%), forest
(77%), and residential (6.8%) land uses (Xing et al., 2013). Elevation in the watershed
ranges from 127 to 432 m above mean sea level (Fig. 2). The soil in the study sites is
classified as mineral, derived from various parent materials. The major associations are
Caribou, Carleton, Glassville, Grandfalls, Holmesville, McGee, Muniac, Siegas, Thibault,
Undine, Victoria, Waasis, and one organic soil (Fig. 3). The study site belongs to the
Upper Saint John River Valley Ecoregion in the Atlantic Maritime Ecozone (Marshall et
al., 1999). The climate of the region is considered to be moderately cool boreal with
approximately 120 frost-free days, annually (Yang et al., 2009). The average temperature
is 3.7˚C and annual precipitation is 1037.4 mm (Zhao et al., 2008). About one-third of the
precipitation is in the form of snow. Snowmelt leads to major surface runoff and
groundwater recharge events from March to May (Chow and Rees, 2006). The land use
and soil maps in the setup of SWAT for LRW were derived from publicly available data
[Energy and Resource Development (ERD), New Brunswick; Fig. 3].




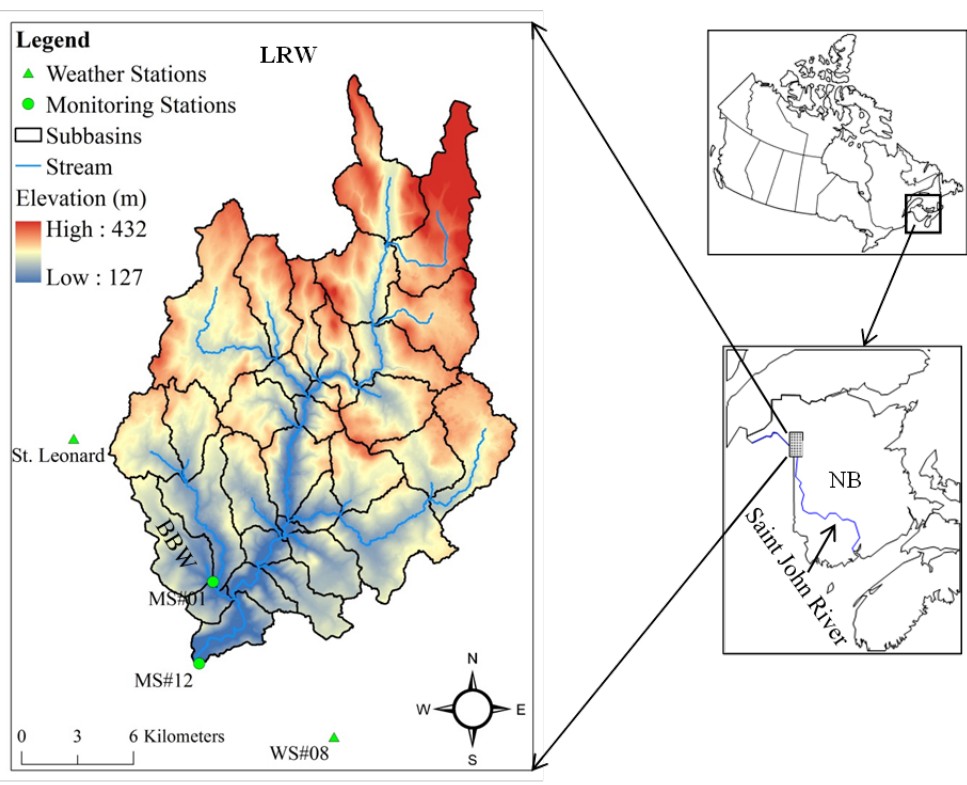


**Fig. 2** Location of the LRW and BBW and water-monitoring stations #01 and #12 as well

as weather stations #08 and St. Leonard. Elevations and subbasins are also shown for

LRW.





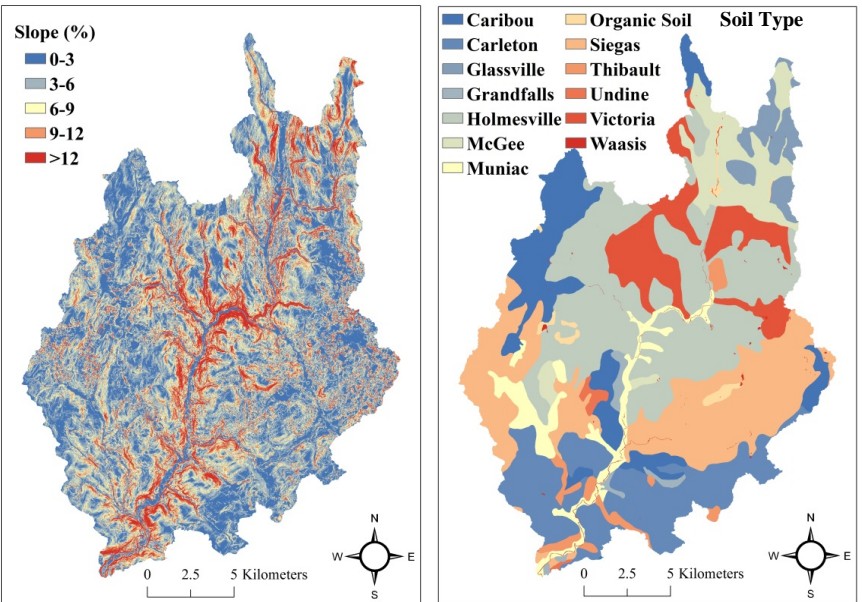


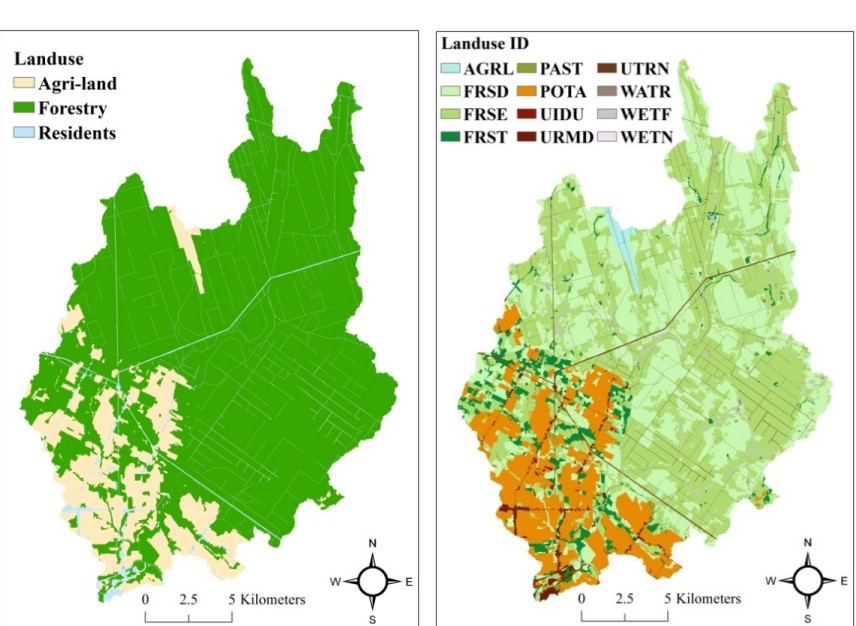

**Fig. 3** Slope classes created using a 10-m resolution LiDAR (Light Detection and
Ranging)-based DEM (Digital Elevation Model), soil and land use maps, and land use
IDs used by SWAT (see Table 2 for land use ID meaning).



The small experimental watershed of the study is the Black Brook Watershed (BBW),
a subbasin of LRW (Fig. 2). The BBW has been studied extensively for more than 20
years to evaluate the impact of agriculture on soil erosion and water quality (Chow and
Rees, 2006; Li et al., 2014). The watershed covers an area of 14.5 km$^2$, with 65% being
agriculture land, 21% forest land, and 14% residential areas and wetlands. Slopes vary
from 1-6% in the upper basin to 4-9% in the central area. In the lower portion of the
watershed, slopes are more strongly rolling at 5-16%. Soil surveys (1:10,000 scale)
identified six mineral soils, namely Grandfalls, Holmesville, Interval, Muniac, Siegas,
and Undine, and one organic soil, St. Quentin (Mellerowicz, 1993).
A water-monitoring station was established at the outlet of BBW in 1992 (MS#01; Fig.
2) and another (MS#12) at the outlet of LRW in 2001. At these stations, V-notch weirs
were installed, and the stage height of the water was recorded using a Campbell-
Scientific CR10X data logger. Stage height values were converted to total flow rates with
a calibration curve function (Chow et al., 2011). Water samples were collected with an
ISCO automatic sampler. Sampling frequency was set at one sample every 72 hours when
runoff was absent. During runoff events, sampling frequency was increased to one
sample every 5-cm change in stage height. Samples were analyzed for concentration of
suspended solids, nitrate-nitrogen ($NO_3$-N), and soluble-phosphorus (Sol-P). Detailed
description of data collection procedures and sample analyses can be found in Chow et al.
(2011). Weather data including daily precipitation, air temperature, relative humidity, and
wind speed were acquired from the St. Leonard Environment Canada weather station,
located approximately 5 km northwest of BBW (Fig. 2). The daily average relative
humidity and wind speed were calculated based on hourly values. Since this weather



station did not monitor daily solar radiation, the study used solar radiation collected from
a weather station located approximately 10 km southeast of BBW (WS#08; Fig. 2).
**2.2 Modification of SWAT**
As a process-based semi-distributed watershed model, SWAT is designed to simulate
hydrological processes and predict water quantity and quality as affected by land use,
land management practices, and climate change (Arnold et al., 1998). It provides a
flexible framework that allows for simulations of the impact of a broad range of BMPs,
such as crop cover, filter strips, conservation tillage, irrigation, and flood-prevention
structures (Gassman et al., 2005; Ullrich and Volk, 2009). The SWAT-model is currently
one of the most commonly used hydrological models to study nonpoint source pollution
problems (Behera and Panda, 2006) and evaluate the impact of BMPs on water quantity
and quality at various spatial scales (Gassman et al., 2005).
Many studies have used SWAT as a decision support tool to evaluate water resources in
large ungauged watersheds. It is believed that SWAT is able to provide reliable
evaluations even without calibration. SWAT analyzes hydrological processes for
watersheds by discretizing them into subbasins, which are then themselves subdivided
into hydrological response units (HRUs) of homogeneous land use, soil properties, and
slope (Yan et al., 2013; Yang et al., 2009). The model calculates the water balance, crop
growth, nutrient cycling, and pesticide movement at the HRU level. Water flow and
sediment and nutrient transport from each HRU are summed and the resulting loadings
are then routed by means of channels, ponds, and reservoirs to the watershed outlet.
Model outputs include HRU-, subbasin-, and watershed-level values of surface, lateral,
and base flows, as well as sediment and nutrient loadings.



In Atlantic Canada, where substantial snow accumulates, SWAT-predicted soil
temperatures have been found to disagree with field measurements (Yang et al., 2009),
especially in winter. To address this discrepancy new physically-based soil-temperature
and snowmelt modules were previously developed for SWAT to account for snow-
insulation effects (Qi et al., 2016a, b) and rain-on-snow events (Qi et al., 2017a). Further
modification to SWAT included a modification to the universal soil loss equation
(MUSLE) by introducing a variable soil erodibility coefficient (K-factor) to address
effects of freeze-thaw cycles on erosion in cold regions (Qi et al. 2017b). The following
changes to SWAT have improved the overall accuracy of the simulations when tested
against field measurements.
**2.3 SWAT Setup, Calibration, and Validation for BBW and LRW**
The new SWAT model has been subsequently set up, calibrated, and validated for
BBW as reported in Qi et al. (2017b). Specific model inputs for both watersheds are
provided in Table 1. The same weather data were used for both watersheds (Table 1). The
Digital Elevation Model (DEM) for LRW and BBW (Qi et al., 2017b) were both based
on high resolution LiDAR (Light Detection and Ranging) data, the first was created at
10-m and the second, at 1-m resolution (Qi et al., 2017b). The LRW was delineated into
32 subbasins from which their topographic characteristics were defined (Fig. 2). The soil
types and slopes, which were classified into five separate classes, are illustrated in Fig. 3
for LRW. After combining the soil, slope, and land use maps through the ArcSWAT-
interface function, 362 HRUs were subsequently created for LRW.






**Table 1** Datasets in SWAT setup, calibration, and validation for BBW and LRW.

| Dataset | BBW | LRW |
|---|---|---|
| LiDAR DEM resolution | 1-m | 10-m |
| Soil map | Survey (1993) | ERD |
| Land use maps | Survey (92-11) | ERD (one map) |
| Precipitation, temperature, relative humidity & wind speed | St. Leonard (92-11) | St. Leonard (01-10) |
| Solar radiation | WS#08 (92-11) | WS#08 (01-10) |
| Contour tillage operation (spring and fall) | Survey (92-11) | Only for potato and barley (01-10) |
| Fertilizer application | Survey (92-11) | Estimated from BBW (2001) |
| Crop rotation | Survey (92-11) | Potato-barley (01-10) |
| Terraces and grassed waterways | Survey (92-11) | Negligible |
| Discharge, sediment, $NO_3$-N and Sol-P | MS#01 (92-11) | MS#12 (01-10) |



Since only one land use map was available for LRW (Table 1), assumptions were
made based on information available on land use and management records for BBW to
adjust the SWAT-management files for LRW as follows:
(1) Potato-barley rotations were assigned to the land use ID POTA (Table 2); for other
land use IDs, a single crop was considered;
(2) Fertilizers were applied only to potato and barley fields, and fertilizer amounts and
N:P (nitrogen-to-phosphorus) ratios were averaged for potato and barley fields over the
entire watershed, based on 2001 survey data from BBW;
(3) Contour tillage was applied only to potato and barley fields;
(4) Flow diversion terraces (FDT) and grassed waterways in LRW were assumed not
used. It is worth noting that these four assumptions serve as a baseline scenario for the
assessment of FDT in LRW at a later time.
In order to evaluate the global performance of the decision support tool for LRW,
related land use and management files were prepared and accessed by SWAT. For





purpose of comparison, simulations with SWAT were produced in an initial application
by setting the adjustable parameters of the model to their default values, and in a second
application by setting the parameters according to values produced with a watershed-
specific model calibration to BBW. This approach with model parameterization is widely
accepted when applying SWAT to large ungauged watersheds (Panagopoulos et al.,

2011).

**2.4 Decision Rules**
The decision support tool was designed to use the "decision rules" to estimate annual
discharge and sediment and nutrient loadings from individual grid cells:

$A = \sum_{i=1}^{n} DR_i \cdot A_i,$            (1)

where $A$ is the annual discharge or sediment and nutrient loadings at the outlet of the
watershed, $DR_i$ and $A_i$ are the delivery ratios and annual discharge or loadings,
respectively, for grid cell $i$. For the present study, statistical equations derived from
simulations of the calibrated version of the enhanced SWAT-model for BBW (Qi et al.,
2017b) were defined as the "decision rules" in the decision support tool.
**2.4.1  Land Use Groups and BMP Scenarios**
In statistical equation development, land use in BBW (24, in total) was first classified
into five land use classes according to their influences on hydrological processes (Table
2). Note that WATR was not used due to its small overall coverage (Fig. 3). As for
watershed management, we considered three main BMPs, i.e.,



(1) FDT + contour tillage;
(2) Contour tillage; and
(3) No-BMP (without FDT and contour tillage).

**Table 2** Land use and land use groups (LUGP) for BBW and LRW.

| LUGP | Land use ID in SWAT | Land use type |
|---|---|---|
| AGRL (General crops) | AGRL | Agricultural Land-Generic |
| | CANA | Canola |
| | CRON | Corn |
| | FPEA | Field peas |
| | POTA | Potato |
| GRAN (Grains) | BARL | Barley |
| | OATS | Oats |
| | PMIL | Millet |
| | RYE | Rye |
| | SWHT | Spring wheat |
| | WWHT | Winter wheat |
| GRAS (Grasses) | BERM | Bermuda grass |
| | CLVR | Clover |
| | HAY | Hay |
| | PAST | Past |
| | RYEG | Ryegrass |
| | TIMO | Timothy |
| FORT (Forestry) | FRSD | Forest-Deciduous |
| | FRSE | Forest-Evergreen |
| | FRST | Forest-Mixed |
| | RNGB | Range-Bush |
| | WETF | Wetlands-Forested |
| | WETN* | Wetlands-No-Forest |
| NOCR (Non-vegetated lands) | URMD | Residential |
| | UTRN | Transportation |
| | UIDU* | Industrial |

Note: "*" indicates unique land use types to LRW not present in BBW and, therefore, unaccounted for in the development of the decision support tool.


The calibrated version of the enhanced SWAT-model for BBW was used to generate
annual outputs based on HRUs from 1992 to 2011. The model was ran three times to
generate the BMP-specific data for statistical equation development.





### 2.4.2 Explanatory Variables Selection

Explanatory candidate variables must be physically-meaningful in hydrological and biochemical processes. It is worth noting that both continuous and categorical variables were included in the regression equation. The land use group (LUGP) was the only categorical variable, and the remaining were all continuous variables. To detect significant predictors, the analysis of covariance (ANCOVA) was used. It requires at least one continuous and one categorical explanatory variable and is used to identify the major and interaction of predictor variables. By including continuous variables, the method can reduce the variance of error to increase the statistical power and precision in estimating categorical variables (Keselman et al., 1998; Li et al., 2014). Inclusion of interaction terms in these regression models dramatically increased model performance.

In the present study, we only considered interactions between two explanatory variables at a time. Student t-tests were conducted to examine the statistical significance of each level of LUGP and their interaction with the various continuous variables. When one level of LUGP (e.g., GRAN; Table 2) did not significantly correlate with water quality or quantity, or there were nominal interactions between a given level and other explanatory variables, this particular level of LUGP would be combined with other levels of LUGP until all new levels of LUGP were statistically significant.

Multiple linear regression analyses were used to relate annual total discharge (mm) and sediment (t ha$^{-1}$), $NO_3$-N (kg ha$^{-1}$), and Sol-P (kg ha$^{-1}$) loadings to the explanatory variables. These work was conducted in R (Ihaka and Gentleman, 1996). Only six continuous explanatory variables were determined for the specification of the statistical models. Annual precipitation (PCP), annual mean air temperature (TMP), and mean





saturated hydraulic conductivity of soil (SOL_K) were common to the dependent
variables (i.e., total discharge and sediment, $NO_3$-N, and Sol-P loadings). The LS-factor
(USLE_LS) and annual N and P application rates (N_APP and P_APP) were unique to
the equations addressing sediment, $NO_3$-N, and Sol-P loading.
**2.4.3   Delivery Ratio Definition**
The LS-factor of the universal soil loss equation (USLE) was determined by slope
gradient (*slp*) and slope length (*L*) of individual HRUs:

$$\text{USLE\_LS} = \left\{\frac{L}{22.1}\right\}^m \cdot (65.41 \cdot sin^2(a) + 4.56 \cdot \sin(a) + 0.065) \qquad (2)$$

where *m* is the equation exponent and *a* is the angle of the slope (in degrees). The
exponent *m* is calculated by,

$$m = 0.6 \cdot (1 - \exp[-35.835 \cdot slp]) \qquad (3)$$

where *slp* is in units of m m$^{-1}$. For the decision support tool, slope length *L* equals to the
length of the grid side and slope gradient was determined by the *Slope* tool in ArcGIS.
The sediment-delivery ratio was not considered in the decision support tool application to
BBW. We assumed that annual sediment loadings from grid cells in decision support tool
were all exported to the outlet of BBW. However, when the decision support tool was
applied to LRW, the sediment-delivery ratio was used to correct estimates of sediment
loading at the watershed outlet. The sediment loadings at the outlet of LRW (*sed*) were
determined by




$sed = SDR \cdot sed\tilde{}$          (4)

where $sed\tilde{}$ is the sediment loading calculated with the sediment loading equation (one for
each BMP and land use group), and *SDR* is determined by (Vanoni, 1975)

$SDR = 0.37 \cdot D^{-0.125}$          (5)

where $D$ (km$^{-2}$) is the drainage area. For annual discharge and nutrient loadings, we
assumed their delivery ratios equal to 1.0 for all grid cells in LRW.
**2.5 Decision Support Tool Assessment (LBAT)**
Inputs to the decision support tool included the six continuous explanatory variables
and LUGP as well as information on management practices, e.g., contour tillage and FDT
implementation. Simulations from each grid cells were summarized at the outlet of the
study watersheds. We first tested the impact of cell size on simulations of water quantity
and quality at the outlet of BBW. The cell size range was determined by considering
different farmland sizes in the watershed. We assumed that farmland-based grid cells can
sufficiently represent basic hydrological processes, land use change, and management
practice implementations for hydrological modeling. Simulated annual water flow and
sediment and nutrient loadings with the decision support tool were compared with those
produced with the calibrated version of the enhanced SWAT-model. Subsequently, the
decision support tool was applied to LRW, and the simulations were compared with the
results of the uncalibrated and calibrated versions of SWAT. The purpose of this was to





test if the decision support tool (i.e., land use and BMP assessment tool; LBAT)
performed better, or at least as well, as both the uncalibrated and calibrated version of
SWAT.
Model performance in terms of water quantity and quality at the outlet of the study
watersheds was assessed based on the coefficient of determination ($R^2$) and relative error
(Re), i.e.,

$$R^2 = \left( \frac{\sum_{i=1}^{n}(O_i - O_{avg}) \cdot (P_i - P_{avg})}{\left[ \sum_{i=1}^{n}(O_i - O_{avg})^2 \cdot \sum_{i=1}^{n}(P_i - P_{avg})^2 \right]^{0.5}} \right)^2 \tag{6}$$

$$Re = \frac{(P_{avg} - O_{avg})}{O_{avg}} \cdot 100\% \tag{7}$$

where $O_i$, $P_i$, $O_{avg}$, and $P_{avg}$ are the observed and predicted and averages of the observed
and predicted values, respectively.
**2.6 FDT Assessment in LRW**
A series of FDT-implementation scenarios were set up for LBAT based on six slope
classes to assess the impact of FDT on water quantity and quality on agricultural lands in
LRW (Fig. 3; Table 3). From scenarios one (S1) to six (S6), total area protected by FDT
gradually increased until all agricultural lands were protected (Table 3). Mean annual
simulations of total discharge and sediment, $NO_3$-N, and Sol-P loadings from LRW from
2001 to 2010 were compared with those of the baseline scenario (FDT = 0%) for each





scenario using two performance indicators, i.e., mean difference (MD) and % relative
difference (PRD), given as:
(1) MD = output with FDT – output without FDT, and
(2) PRD (%) = MD/output without FDT × 100.

(3)

**Table 3** Slope classes and corresponding areas in the agricultural land of LRW.

| Scenario | Slope | Area protected by FDT (ha) | Agricultural lands (%) |
|----------|-------|-----------------------------|------------------------|
| S1 | ≥5% | 624 | 10 |
| S2 | ≥4% | 1328 | 22 |
| S3 | ≥3% | 2224 | 37 |
| S4 | ≥2% | 3680 | 61 |
| S5 | ≥1% | 5360 | 89 |
| S6 | ≥0 | 6048 | 100 |


**3.  Results and Discussion**
**3.1  Statistical Equations (Decision Rules)**
**3.1.1  Model Structure and Coefficients**
Linear regression equations and their explanatory variables for annual discharge and
sediment, $NO_3$-N, and Sol-P loadings under different combinations of land use groups
and BMP scenarios are provided in Tables 4 and 5. In total, three discharge models (Dis1,
Dis2, and Dis3) and five sediment (Sed1_1, Sed1_2, Sed1_3, Sed2, and Sed3), $NO_3$-N
(N1_1, N1_2, N1_3, N2, and N3), and Sol-P (P1_1, P1_2, P1_3, P2, and P3) loading
models were developed. Data transformations (via logarithm and power transformations)
were applied to sediment, $NO_3$-N, and Sol-P loadings to meet the assumption of
normality in multiple regression analysis (Table 4). The contour tillage and FDT were
applied only to agricultural lands, including land use groups AGRL, GRAN, and GRAS





(Table 4). For the no-BMP scenario, three separate sediment, $NO_3$-N, and Sol-P loading
models were developed for agricultural lands (AGRL, GRAN, and GRAS), non-
vegetated lands (NOCR), and forest lands (FORT), and one discharge model (Dis1) for
all land use groups (Table 4). It is worth noting that the sediment loading model, Sed3,
was a modified version of Sed1_1 (multiplied by TERR_P) for the FDT + contour tillage
scenario (Table 4), and the values of TERR_P (Qi et al., 2017b) used for Sed3 were the
same as the calibrated values in SWAT for BBW (Qi et al., 2017b). Also, $NO_3$-N and
Sol-P loadings (N1_2 and P1_2) for non-vegetated lands (NOCR) were determined as
constants, which were equal to the calculated means of $NO_3$-N and Sol-P loadings
determined by SWAT (i.e., 24 and 0.61 kg ha$^{-1}$, respectively; Table 4).
As for LUGP (including AGRL, GRAN, GRAS, FORT, and NOCR; Table 2), three
new land use groups (i.e., LUGP1, LUGP2, and LUGP3) were formulated by combining
agricultural lands AGRL, GRAN, and GRAS during model development (Tables 4 and 5).
For example, LUGP2 was derived by combining AGRL, GRAN, and GRAS on total
discharge (i.e., Dis1 model). Individual model structures are shown in Table 4, whereas
the explanatory variables for these models appear in Tables 6, 7, 8 and 9. The coefficients
estimated for the explanatory variables and their interactions, and their t-test results are
also shown. Most of the $p$-values for these explanatory variables were < 0.001, except for
several that were between 0.001 and 0.08, which were also taken as acceptable.





**Table 4** Statistical models based on land use groups (LUGP) and BMPs.

| BMPs | LUGP* | Model | Structure |
|---|---|---|---|
| No-BMP | CRGP2,NOCR,FORT | Dis1 | Discharge = $f$ (PCP, TMP, SOL_K, LUGP2) |
| Contour tillage | AGRL,GRAN,GRAS | Dis2 | = $f$ (PCP, TMP, SOL_K) |
| FDT+Contour tillage | AGRL,GRAN,GRAS | Dis3 | = $f$ (PCP, TMP, SOL_K) |
| No-BMP | CRGP1,GRAS | Sed1_1 | Sediment$^{(1/10)}$ = $f$ (USLE_LS, PCP, TMP, SOL_K, LUGP1) |
|  | NOCR | Sed1_2 | = $f$ (USLE_LS, PCP) |
|  | FORT | Sed1_3 | = $f$ (USLE_LS, PCP, SOL_K) |
| Contour tillage | CRGP1,GRAS | Sed2 | Sediment$^{(1/10)}$ = $f$ (USLE_LS, PCP, TMP, SOL_K, LUGP1) |
| FDT+Contour tillage | AGRL,GRAN,GRAS | Sed3 | Sediment = Sed1_1 × TERR_P |
| No-BMP | AGRL,GRAN,GRAS | N1_1 | Log($NO_3$-N) = $f$ (N_APP, PCP, TMP, SOL_K, LUGP) |
|  | NOCR | N1_2** | $NO_3$-N= 24 kg ha$^{-1}$ |
|  | FORT | N1_3 | Log($NO_3$-N) = $f$ (PCP, TMP, SOL_K) |
| Contour tillage | AGRL,GRAN,GRAS | N2 | Log($NO_3$-N) = $f$ (N_APP, PCP, TMP, SOL_K, LUGP) |
| FDT+Contour tillage | CRGP3,GRAN | N3 | = $f$ (N_APP, PCP, TMP, SOL_K, LUGP3) |
| No-BMP | CRGP1,GRAS | P1_1 | Log(Sol-P) = $f$ (P_APP, PCP, TMP, SOL_K, LUGP1) |
|  | NOCR | P1_2** | Sol-P = 0.61 kg ha$^{-1}$ |
|  | FORT | P1_3 | Log(Sol-P) = $f$ (PCP, TMP, SOL_K) |
| Contour tillage | CRGP1,GRAS | P2 | Log(Sol-P) = $f$ (P_APP, PCP, TMP, SOL_K, LUGP1) |
| FDT+Contour tillage | AGRL,GRAN,GRAS | P3 | = $f$ (P_APP, PCP, TMP, SOL_K, LUGP) |

* AGRL and GRAN are combined into one group, namely CRGP1 in LUGP1; AGRL, GRAN and GRAS are combined into one group, namely
CRGP2 in LUGP2; AGRL and GRAN are combined into one group, namely CRGP1 in LUGP1; AGRL, GRAN and GRAS are combined into one group, namely CRGP3 in LUGP3; ** variable is set constant.







**Table 5** Explanatory variables determined for statistical analysis.

| Variable | Unit | Meaning |
| --- | --- | --- |
| LUGP | — | Land use groups including AGRL, GRAN, GRAS, FORT, and NOCR |
| LUGP1 | — | AGRL and GRAN are combined into a new group, CRGP1 |
| LUGP2 | — | AGRL, GRAN, and GRAS are combined into a new group, CRGP2 |
| LUGP3 | — | AGRL and GRAS are combined into a new group, CRGP3 |
| N_APP | kg ha$^{-1}$ | Annual N application rate |
| P_APP | kg ha$^{-1}$ | Annual P application rate |
| PCP | mm | Annual precipitation |
| SOL_K | mm h$^{-1}$ | Mean saturated hydraulic conductivity of soil |
| TERR_P | — | P-factor for FDT |
| TMP | ℃ | Annual mean air temperature |
| USLE_LS | — | LS-factor of USLE |






















**Table 6** Coefficient values for the three discharge models corresponding to land use and

BMPs described in Table 4.

| Model variable | Estimate | Std. Error | t-value | *p*-value |
|---|---|---|---|---|
| **Dis1** | | | | |
| Intercept | -1565 | 24.04 | -65.089 | <0.001 |
| PCP | 1.933 | 0.02176 | 88.837 | <0.001 |
| TMP | 282.7 | 6.091 | 46.402 | <0.001 |
| SOL_K | 0.06338 | 0.00992 | 6.389 | <0.001 |
| FORT | 30.79 | 14.16 | 2.175 | 0.030 |
| NOCR | 162.2 | 14.51 | 11.181 | <0.001 |
| PCP:TMP | -0.2488 | 0.005487 | -45.352 | <0.001 |
| PCP:FORT | 0.04684 | 0.01191 | 3.934 | <0.001 |
| PCP:NOCR | -0.0535 | 0.01224 | -4.37 | <0.001 |
| TMP:FORT | 9.723 | 1.684 | 5.775 | <0.001 |
| TMP:NOCR | 4.506 | 1.731 | 2.603 | 0.009 |
| SOL_K:FORT | -0.3769 | 0.03403 | -11.076 | <0.001 |
| SOL_K:NOCR | -0.2959 | 0.032 | -9.248 | <0.001 |
| **Dis2** | | | | |
| Intercept | -1633 | 27.29 | -59.84 | <0.001 |
| PCP | 1.995 | 0.02472 | 80.69 | <0.001 |
| TMP | 302.2 | 6.87 | 43.98 | <0.001 |
| SOL_K | 0.08696 | 0.01167 | 7.45 | <0.001 |
| PCP:TMP | -0.2662 | 0.006199 | -42.94 | <0.001 |
| **Dis3** | | | | |
| Intercept | -1666 | 36.58 | -45.54 | <0.001 |
| PCP | 2.007 | 0.03305 | 60.713 | <0.001 |
| TMP | 298 | 9.351 | 31.865 | <0.001 |
| SOL_K | 0.09353 | 0.01573 | 5.946 | <0.001 |
| PCP:TMP | -0.2606 | 0.008406 | -31.004 | <0.001 |









**Table 7** Coefficient values for the four sediment loading models corresponding to land
use and BMPs described in Table 4.

| Model variable | Estimate | Std. Error | t-value | *p*-value |
|---|---|---|---|---|
| **Sed1_1** | | | | |
| Intercept | 0.2749 | 0.06125 | 4.488 | <0.001 |
| USLE_LS | 0.1201 | 0.02224 | 54.018 | <0.001 |
| PCP | 0.000788 | 5.54E-05 | 14.218 | <0.001 |
| TMP | 0.1117 | 0.01528 | 7.307 | <0.001 |
| SOL_K | 0.000568 | 0.00022 | 2.585 | 0.010 |
| GRAS | -0.0353 | 0.00881 | -4.007 | <0.001 |
| USLE_LS:SOL_K | -0.00014 | 4.69E-05 | -3.045 | 0.002 |
| USLE_LS:GRAS | -0.02623 | 0.006826 | -3.842 | <0.001 |
| PCP:TMP | -0.00011 | 1.38E-05 | -7.967 | <0.001 |
| PCP:SOL_K | -4.6E-07 | 1.91E-07 | -2.406 | 0.016 |
| **Sed1_2** | | | | |
| Intercept | 0.8575 | 0.008826 | 97.15 | <0.001 |
| PCP | 0.000123 | 7.82E-06 | 15.67 | <0.001 |
| PCP:USLE_LS | 0.000209 | 5.02E-06 | 41.65 | <0.001 |
| **Sed1_3** | | | | |
| (Intercept) | 0.3992 | 0.02267 | 17.613 | <0.001 |
| USLE_LS | 0.07935 | 0.01967 | 4.034 | <0.001 |
| PCP | 0.000204 | 1.96E-05 | 10.371 | <0.001 |
| SOL_K | 0.000545 | 5.71E-05 | 9.534 | <0.001 |
| USLE_LS:PCP | 4.94E-05 | 1.71E-05 | 2.9 | 0.004 |
| USLE_LS:SOL_K | -0.00067 | 4.89E-05 | -13.718 | <0.001 |
| **Sed2** | | | | |
| Intercept | 0.2591 | 0.05228 | 4.956 | <0.001 |
| USLE_LS | 0.12 | 0.001898 | 63.218 | <0.001 |
| PCP | 0.000767 | 4.73E-05 | 16.212 | <0.001 |
| TMP | 0.1162 | 0.01304 | 8.907 | <0.001 |
| SOL_K | 0.000746 | 0.000188 | 3.981 | <0.001 |
| GRAS | -0.06937 | 0.01648 | -4.211 | <0.001 |
| USLE_LS:SOL_K | -0.00013 | 4E-05 | -3.137 | 0.002 |
| USLE_LS:GRAS | -0.02662 | 0.005829 | -4.567 | <0.001 |
| PCP:TMP | -0.00011 | 1.18E-05 | -9.522 | <0.001 |
| PCP:SOL_K | -6.3E-07 | 1.63E-07 | -3.846 | <0.001 |
| TMP:GRAS | 0.007415 | 0.003664 | 2.024 | 0.043 |






**Table 8** Coefficient values for the four NO$_3$-N loading models corresponding to land use

and BMPs described in Table 4.

| Model variable | Estimate | Std. Error | t-value | p-value |
|---|---|---|---|---|
| **N1_1** | | | | |
| Intercept | 1.44 | 0.1753 | 8.213 | <0.001 |
| N_APP | -0.00862 | 0.000699 | -12.325 | <0.001 |
| PCP | 0.000543 | 0.00016 | 3.4 | <0.001 |
| TMP | 0.1363 | 0.03357 | 4.059 | <0.001 |
| SOL_K | -0.00344 | 9.78E-05 | -35.163 | <0.001 |
| GRAN | -1.117 | 0.1021 | -10.937 | <0.001 |
| GRAS | -1.97 | 0.1562 | -12.611 | <0.001 |
| N_APP:PCP | 5.31E-06 | 6.45E-07 | 8.233 | <0.001 |
| N_APP:TMP | 0.000963 | 7.45E-05 | 12.929 | <0.001 |
| N_APP:SOL_K | 9.6E-06 | 6.4E-07 | 15.024 | <0.001 |
| PCP:GRAN | 0.000677 | 9.38E-05 | 7.215 | <0.001 |
| PCP:GRAS | 0.001029 | 0.000143 | 7.201 | <0.001 |
| PCP:TMP | -0.00025 | 2.64E-05 | -9.467 | <0.001 |
| TMP:GRAN | 0.1 | 0.01134 | 8.817 | <0.001 |
| TMP:GRAS | 0.2132 | 0.01651 | 12.912 | <0.001 |
| **N1_3** | | | | |
| Intercept | -1.411 | 0.3087 | -4.573 | <0.001 |
| PCP | 0.001875 | 0.000279 | 6.710 | <0.001 |
| TMP | 0.4437 | 0.07831 | 5.666 | <0.001 |
| SOL_K | -0.00104 | 0.000116 | -8.979 | <0.001 |
| PCP:TMP | -0.00032 | 7.06E-05 | -4.484 | <0.001 |
| **N2** | | | | |
| Intercept | 1.429 | 0.1757 | 8.134 | <0.001 |
| N_APP | -0.00858 | 0.000701 | -12.233 | <0.001 |
| PCP | 0.000548 | 0.00016 | 3.425 | <0.001 |
| TMP | 0.1376 | 0.03365 | 4.089 | <0.001 |
| SOL_K | -0.00345 | 9.8E-05 | -35.223 | <0.001 |
| GRAN | -1.11 | 0.1023 | -10.849 | <0.001 |
| GRAS | -1.962 | 0.1566 | -12.526 | <0.001 |
| N_APP:PCP | 5.3E-06 | 6.47E-07 | 8.187 | <0.001 |
| N_APP:TMP | 0.000957 | 7.46E-05 | 12.82 | <0.001 |
| N_APP:SOL_K | 9.65E-06 | 6.4E-07 | 15.067 | <0.001 |
| PCP:GRAN | 0.000674 | 9.41E-05 | 7.167 | <0.001 |
| PCP:GRAS | 0.001026 | 0.000143 | 7.162 | <0.001 |
| PCP:TMP | -0.00025 | 2.64E-05 | -9.456 | <0.001 |





| | | | | |
|---|---|---|---|---|
| TMP:GRAN | 0.09934 | 0.01137 | 8.738 | <0.001 |
| TMP:GRAS | 0.2122 | 0.01655 | 12.821 | <0.001 |
| **N3** | | | | |
| Intercept | -0.3595 | 0.1718 | -2.092 | 0.037 |
| N_APP | -0.00131 | 0.000435 | -3.011 | 0.003 |
| PCP | 0.001621 | 0.00015 | 10.806 | <0.001 |
| TMP | 0.3977 | 0.03857 | 10.312 | <0.001 |
| SOL_K | -0.00386 | 0.000505 | -7.641 | <0.001 |
| GRAN | -0.2133 | 0.07504 | -2.842 | 0.005 |
| N_APP:PCP | 1.65E-06 | 3.59E-07 | 4.61 | <0.001 |
| N_APP:TMP | 0.000281 | 4.74E-05 | 5.939 | <0.001 |
| N_APP:GRAN | 0.000716 | 0.000292 | 2.453 | 0.014 |
| PCP:TMP | -0.00035 | 3.32E-05 | -10.506 | <0.001 |
| PCP:SOL_K | 1.21E-06 | 4.36E-07 | 2.781 | 0.005 |
| PCP:GRAN | 0.000267 | 5.82E-05 | 4.577 | <0.001 |
| TMP:GRAN | -0.04685 | 0.008004 | -5.853 | <0.001 |




















**Table 9** Coefficient values for four Sol-P models corresponding to land use and BMPs
described in Table 4.

| Model variable | Estimate | Std. Error | t-value | *p*-value |
|---|---|---|---|---|
| **P1_1** | | | | |
| Intercept | -3.711 | 0.1306 | -28.416 | <0.001 |
| P_APP | 0.002341 | 0.000623 | 3.757 | <0.001 |
| PCP | 0.003195 | 0.000117 | 27.286 | <0.001 |
| TMP | 0.5542 | 0.03197 | 17.337 | <0.001 |
| SOL_K | 0.00298 | 0.000472 | 6.305 | <0.001 |
| GRAS | -0.4321 | 0.0382 | -11.312 | <0.001 |
| P_APP:PCP | -2.4E-06 | 5.2E-07 | -4.64 | <0.001 |
| P_APP:TMP | 0.000829 | 7.7E-05 | 10.797 | <0.001 |
| PCP:TMP | -0.00052 | 2.9E-05 | -18.297 | <0.001 |
| PCP:SOL_K | -1.2E-06 | 3. 97E-07 | -3.095 | 0.002 |
| TMP:SOL_K | -0.00026 | 5.7E-05 | -4.526 | <0.001 |
| TMP:GRAS | 0.03787 | 0.00941 | 4.024 | <0.001 |
| **P1_3** | | | | |
| Intercept | -4.43817 | 0.589848 | -7.512 | <0.001 |
| PCP | 0.002509 | 0.000534 | 4.701 | <0.001 |
| TMP | 0.417306 | 0.1496445 | 2.789 | 0.005 |
| SOL_K | 0.001247 | 0.000222 | 5.622 | <0.001 |
| PCP:TMP | -0.0003 | 0.000135 | -2.253 | 0.024 |
| **P2** | | | | |
| Intercept | -3.667 | 0.1357 | -27.017 | <0.001 |
| P_APP | 0.003461 | 0.000663 | 5.218 | <0.001 |
| PCP | 0.003017 | 0.000122 | 24.783 | <0.001 |
| TMP | 0.5149 | 0.03304 | 15.584 | <0.001 |
| SOL_K | 0.003531 | 0.000488 | 7.233 | <0.001 |
| GRAS | -0.2039 | 0.09001 | -2.265 | 0.024 |
| P_APP:PCP | -2.4E-06 | 5.54E-07 | -4.305 | <0.001 |
| P_APP:TMP | 0.000432 | 7.93E-05 | 5.445 | <0.001 |
| P_APP:GRAS | -0.03304 | 0.007019 | -4.707 | <0.001 |
| PCP:TMP | -0.00044 | 2.95E-05 | -14.952 | <0.001 |
| PCP:SOL_K | -1.4E-06 | 4.1E-07 | -3.446 | <0.001 |
| PCP:GRAS | -0.00025 | 7.66E-05 | -3.25 | 0.001 |
| TMP:SOL_K | -0.00025 | 5.87E-05 | -4.184 | <0.001 |
| TMP:GRAS | 0.05117 | 0.009839 | 5.201 | <0.001 |
| **P3** | | | | |
| Intercept | -2.817 | 0.2548 | -11.054 | <0.001 |
| P_APP | -0.01363 | 0.001854 | -7.352 | <0.001 |
| PCP | 0.002778 | 0.000178 | 15.609 | <0.001 |





| | | | | |
|---|---|---|---|---|
| TMP | 0.1406 | 0.06523 | 2.155 | 0.031 |
| SOL_K | 0.00651 | 0.000702 | 9.279 | <0.001 |
| GRAN | -0.9386 | 0.1378 | -6.812 | <0.001 |
| GRAS | -0.9931 | 0.1813 | -5.478 | <0.001 |
| P_APP:TMP | 0.003562 | 0.000491 | 7.252 | <0.001 |
| P_APP:GRAN | 0.007736 | 0.002179 | 3.549 | <0.001 |
| P_APP:GRAS | -0.05489 | 0.01295 | -4.24 | <0.001 |
| PCP:TMP | -0.0003 | 4.42E-05 | -6.763 | <0.001 |
| PCP:SOL_K | -3.7E-06 | 5.78E-07 | -6.359 | <0.001 |
| PCP:GRAN | 0.000112 | 5.1E-05 | 2.192 | 0.028 |
| PCP:GRAS | -0.00019 | 0.000109 | -1.74 | 0.082 |
| TMP:SOL_K | -0.00021 | 8.8E-05 | -2.4 | 0.016 |
| TMP:GRAN | 0.1798 | 0.03332 | 5.397 | <0.001 |
| TMP:GRAS | 0.247 | 0.03581 | 6.898 | <0.001 |


### 3.1.2   Statistical Equation Assessment

Simulations based on the statistical equations and the calculated outputs from
individual HRUs for the different BMPs are compared in Table 10. In general, discharge
models were able to reproduce SWAT simulations for the three BMPs; $R^2$ ranging from
0.86 to 0.9. Mean discharge simulated with the statistical equations was equal to that of
SWAT (Table 10). Mean discharge (636 mm) for the no-BMP-case (BMP 3) was greater
than that for BMPs using contour tillage and FDTs (619 and 628 mm for BMP 1 and 2,
respectively), suggesting that contour tillage and FDTs can cause evapotranspiration to
increase.
Models Sed1_2 and Sed1_3 were able to reproduce simulations with SWAT (yielding
$R^2 = 0.71$ and 0.57, respectively), and simulated mean sediment loadings were close to
that of SWAT (Table 10). Models Sed1_1 and Sed2 tended to underestimate results from
SWAT (Table 10), with an overall lower mean sediment loading of 10.78 vs. 12.84 and
8.31 vs. 9.4 t ha$^{-1}$, respectively. Mean sediment loading with Sed3 (0.89 t ha$^{-1}$) was





slightly greater than that of SWAT (0.84 t ha$^{-1}$), due to the fact that Sed3 only took into
account TERR_P, whereas SWAT took into account TERR_CN and the impact of
grassed waterways. Results from the statistical equations showed that the mean sediment
loading for BMP 2 (8.31 t ha$^{-1}$) was significantly different than that for BMPs 1 and 3,
with mean loading of 0.89 and 10.78 t ha$^{-1}$ (Table 10). The smallest mean sediment
loading (0.09 t ha$^{-1}$) was found to occur with the FORT land use grouping (Table 10).

The four NO$_3$-N and Sol-P loading equations explained ~50% of the variation in the

SWAT simulations for the same variables, with R$^2$ ranging from 0.33 to 0.59 (Table 10).
Mean NO$_3$-N and Sol-P loadings with the statistical equations were all slightly less than
the values produced with SWAT for the different BMPs (Table 10). Mean NO$_3$-N
loadings were greater for BMP 1 (44 kg ha$^{-1}$) than those for BMPs 2 and 3 with both
giving 39 kg ha$^{-1}$ (Table 10), due to increased infiltration with FDT. Mean Sol-P loading
(0.8 kg ha$^{-1}$) was less for BMP 3 than for BMP 2 (0.89 kg ha$^{-1}$), whereas much greater
than for BMP 1 (0.43 kg ha$^{-1}$). Although contour tillage can help reduce sediment loading
by modifying micro-topography and reducing erosion runoff (the reason we set USLE_P
< 1), Sol-P transported with surface runoff increased due to reduced residue cover
protecting the soil surface during winter and during the snowmelt season. When FDT was
implemented with tillage, however, less surface runoff was generated due to increased
infiltration leading to a reduction in Sol-P loading. Mean NO$_3$-N and Sol-P loadings for
the FORT land grouping (10 vs. 0.06 kg ha$^{-1}$) were much less than those of the CRGP
land grouping, 39 vs. 0.8 kg ha$^{-1}$ (Table 10).


**Table 10** Comparisons of simulations of statistical models and outputs from SWAT for different land use groups and BMPs based on mean and standard deviation for the entire simulation period (1992-2011).

| Variable | Index | No-BMP CRGP SWAT | No-BMP CRGP Fitted | No-BMP NOCR SWAT | No-BMP NOCR Fitted | No-BMP FORT SWAT | No-BMP FORT Fitted | Tillage CRGP SWAT | Tillage CRGP Fitted | FDT + Tillage CRGP SWAT | FDT + Tillage CRGP Fitted |
|---|---|---|---|---|---|---|---|---|---|---|---|
| Discharge (mm) | Mean | ↑ | ↑ | 636 | 636 | ↓ | ↓ | 619 | 619 | 628 | 628 |
| | SD | ↑ | ↑ | 144 | 133 | ↓ | ↓ | 140 | 132 | 151 | 143 |
| | $R^2$ | ↑ | ↑ | 0.86 (Dis1) | | ↓ | ↓ | 0.88 (Dis2) | | 0.90 (Dis3) | |
| Sediment (t ha⁻¹) | Mean | 12.84 | 10.78 | 1.80 | 1.71 | 0.10 | 0.09 | 9.40 | 8.31 | 0.84 | 0.89 |
| | SD | 11.86 | 9.44 | 1.94 | 1.95 | 0.14 | 0.16 | 8.28 | 7.38 | 2.72 | 1.18 |
| | $R^2$ | 0.48 (Sed1_1) | | 0.71 (Sed1_2) | | 0.57 (Sed1_3) | | 0.56 (Sed2) | | — | |
| NO₃-N (kg ha⁻¹) | Mean | 43 | 39 | 24 | — | 10 | 10 | 43 | 39 | 47 | 44 |
| | SD | 24 | 14 | 16 | — | 6 | 3 | 24 | 14 | 29 | 21 |
| | $R^2$ | 0.40 (N1_1) | | — | | 0.33 (N1_3) | | 0.39 (N2) | | 0.59 (N3) | |
| Sol-P (kg ha⁻¹) | Mean | 0.88 | 0.80 | 0.61 | — | 0.08 | 0.06 | 0.98 | 0.89 | 0.49 | 0.43 |
| | SD | 0.49 | 0.32 | 0.46 | — | 0.06 | 0.03 | 0.59 | 0.38 | 0.33 | 0.23 |
| | $R^2$ | 0.47 (P1_1) | | — | | 0.38 (P1_3) | | 0.48 (P2) | | 0.52 (P3) | |

Note: CRGP refers to crop groups including AGRL, GRAN, and GRAS; the statistics for discharge in no-BMP scenario are based on CRGP, NOCR, and FORT.





### 3.2 LBAT Assessment

### 3.2.1 Impact of Grid Cell Size on LBAT Simulation

Simulations of water quantity and quality by LBAT with different grid-cell sizes (i.e., 25, 50, 100, 200, and 400 m) for BBW are shown in Fig. 4. Statistical tests indicated that grid-cell size had a significant effect on sediment loading ($p$-value $< 0.01$), with no effect observed for discharge and $NO_3$-N and Sol-P loadings ($p$-values $> 0.99$). Increasing cell size (i.e., slope length) increased sediment loading. However, the mean slope gradient was reduced. As a result, the mean sediment loadings were correlated non-linearly with cell size (Fig. 13). The highest mean sediment loading was found with a cell size of 100 m (5.86 t ha$^{-1}$), whereas the lowest was found to occur with a cell size of 25 and 400 m (3.37 t ha$^{-1}$). The LBAT with a cell size of 25 and 400 m was able to generate sediment loadings consistent with field measurements. Considering computational efficiency, we chose a grid-cell size of 400 m as the basic LBAT-simulation unit for LRW.





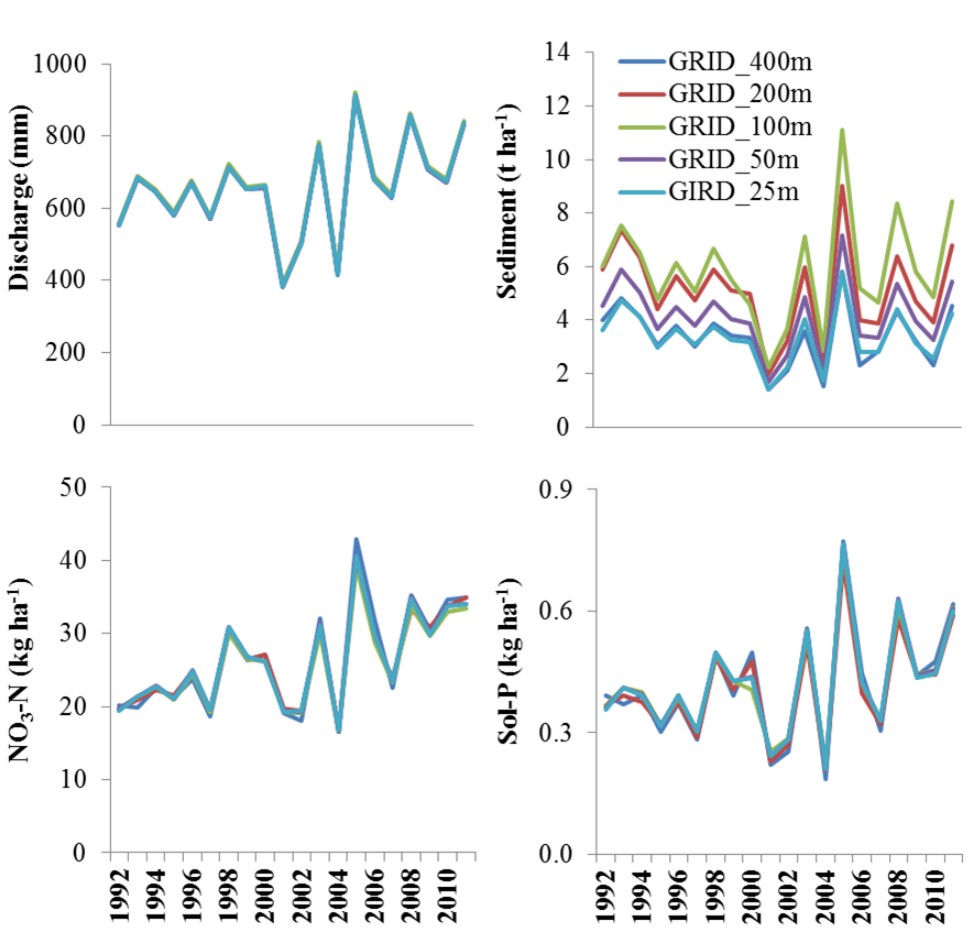

**Fig. 4** LBAT-produced simulations of annual stream discharge and sediment, $NO_3$-N, and

Sol-P loadings determined for different DEM grid-cell sizes (i.e., 25, 50, 100, 200, and

400 m).





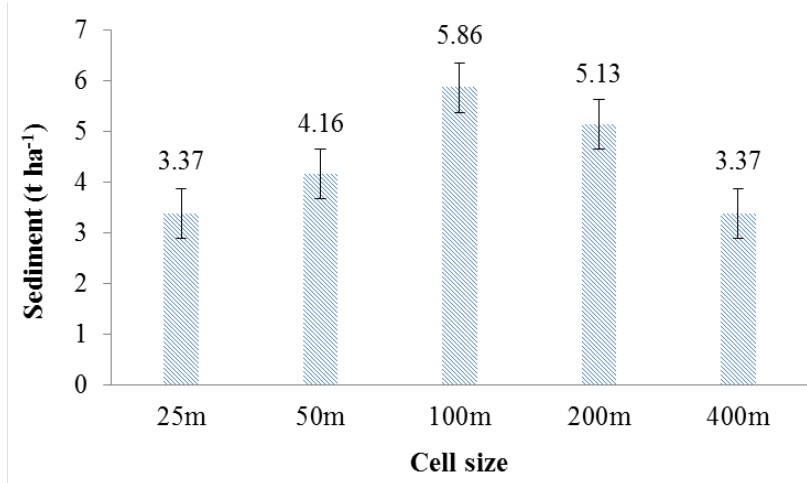


**Fig. 5** Impact of grid-cell size on LBAT-simulation of sediment loading. Mean annual
sediment loadings and standard errors (vertical bars) from 1992 to 2011 are indicated.




### 3.2.2 LBAT vs. SWAT Applications to BBW


Simulations of water quantity and quality with LBAT and SWAT are compared with
field measurements from BBW (Fig. 6). Model assessments are shown in Table 11. Both
LBAT and SWAT were able to capture a significant portion of the variation in measured
annual stream discharge ($R^2$ = 0.48 and 0.56, respectively) and $NO_3$-N and Sol-P
loadings ($R^2$ = 0.25, 0.32, 0.23, and 0.38, respectively); however, this was not the case
when annual sediment loading was considered (Table 11; Fig. 6) due to the fact that the
current version of SWAT does not address soil erosion caused by freeze-thaw cycles (Qi
et al., 2017b). Absolute values of Re with LBAT were less than 48 for these four
variables (Table 11). The mean discharge and sediment loading with LBAT were slightly
less than those of SWAT and field measurements, while the mean Sol-P loading (0.5 kg
ha$^{-1}$) was greater; 0.33 and 0.34 kg ha$^{-1}$ for SWAT and field measurements, respectively
(Table 11). The mean $NO_3$-N loading (30 kg ha$^{-1}$) with LBAT was equal to the mean
based on field measurements, whereas it was slightly greater than that of SWAT (29 kg
ha$^{-1}$). These results indicated that LBAT and SWAT performed equally well in
reproducing estimates of water quantity and quality at the outlet of BBW.




**Fig. 6** Simulations of annual stream discharge and sediment, NO$_3$-N, and Sol-P loadings
with LBAT and SWAT compared with field measurements at the outlet of BBW.




**Table 11** Statistical assessments of LBAT and SWAT in simulations of annual stream
discharge and sediment, NO$_3$-N, and Sol-P loadings at the outlet of BBW for the
simulation period of 1992-2011.

| Variable | Index | Measured | SWAT | LBAT |
|---|---|---|---|---|
| Discharge (mm) | Mean | 696 | 706 | 655 |
| | Re (%) | — | 2 | -6 |
| | $R^2$ | — | 0.56 | 0.48 |
| Sediment (t ha$^{-1}$) | Mean | 3.77 | 3.34 | 3.31 |
| | Re (%) | — | -12 | -12 |
| | $R^2$ | — | 0.02 | 0.02 |
| NO$_3$-N (kg ha$^{-1}$) | Mean | 30 | 29 | 30 |
| | Re (%) | — | -3 | 0 |
| | $R^2$ | — | 0.32 | 0.25 |
| Sol-P (kg ha$^{-1}$) | Mean | 0.34 | 0.33 | 0.50 |
| | Re (%) | — | -3 | 48 |
| | $R^2$ | — | 0.38 | 0.23 |



**3.2.3   LBAT vs. SWAT in LRW**
Simulations of water quantity and quality with LBAT and the uncalibrated and
calibrated versions of SWAT are compared with field measurements for LRW (Fig. 7).
Model assessments for different simulation periods (depending on measurement
availability) are shown in Table 12. It is worth noting that, to eliminate unrealistic results,
USLE_LS was constrained in Sed1_2 to the NOCR land use group:

$$\text{USLE\_LS} = \begin{cases} Eq.\,6\text{-}1 & USLE\_LS \leq 1.28 \\ 1.28 & USLE\_LS > 1.28 \end{cases} \qquad (8)$$




where 1.28 is the maximum USLE_LS for BBW.
In general, the two versions of SWAT and LBAT slightly underestimated annual
stream discharge, capturing its variation reasonably well (Fig. 7a). The uncalibrated and
calibrated versions of SWAT had the least and largest absolute values of Re (Re = -2 and
-9), whereas LBAT Re = -6 (Table 12). The uncalibrated version of SWAT severely
overestimated annual sediment and $NO_3$-N loading (Re = 212 and 87, respectively; Figs.
7b and c), whereas the calibrated version of SWAT and LBAT underestimated sediment
loading (Re = -32 and -52, respectively) and overestimated $NO_3$-N loading (Re = 22 and
27, respectively; Table 12). In general, the calibrated version of SWAT and LBAT
captured the variation in annual sediment and $NO_3$-N loadings reasonably well (Figs. 7b
and c). However, the two versions of SWAT and LBAT failed to capture the variation in
annual Sol-P loadings (Fig. 7d). The LBAT had the smallest absolute value of Re (i.e., Re
= -16), while the uncalibrated and calibrated versions of SWAT had larger values (Re = -
59 and -55, respectively). These results suggested that the LBAT and the calibrated
version of SWAT performed equally well in simulating annual stream flow and sediment
and $NO_3$-N loadings, with LBAT performing slightly better for annual Sol-P loading.
LBAT performed noticably better than the uncalibrated version of SWAT, especially for
annual sediment and $NO_3$-N loadings.







**Table 12** Statistical assessments of LBAT and SWAT for annual stream discharge and
sediment, $NO_3$-N, and Sol-P loadings at the outlet of LRW for different simulation
periods

| Period | Variable | Index | Measurement | SWAT -Uncalibrated | SWAT -Calibrated | LBAT |
|--------|----------|-------|-------------|--------------------|------------------|------|
| 01-07 | Discharge (mm) | Mean | 704 | 691 | 638 | 604 |
| | | Re (%) | — | -2 | -9 | -6 |
| 01-10 | Sediment ($t\ ha^{-1}$) | Mean | 0.95 | 2.95 | 0.65 | 0.45 |
| | | Re (%) | — | 212 | -32 | -52 |
| 03-10 | $NO_3$-N ($kg\ ha^{-1}$) | Mean | 12 | 22 | 14 | 15 |
| | | Re (%) | — | 87 | 22 | 27 |
| 03-10 | Sol-P ($kg\ ha^{-1}$) | Mean | 0.31 | 0.13 | 0.14 | 0.26 |
| | | Re (%) | — | -59 | -55 | -16 |


Since LBAT is based on decision rules (statistical equations) which were derived from
SWAT simulations for BBW, its usage should be constrained to areas with soil,
landscape, and land use characteristics similar to BBW. Input characteristics exceeding
the range of SWAT data considered could lead to large errors in predictions. LBAT is
flexible in its structure, and with thoughtful development of internal rules, it can be
applied to diverse environments.





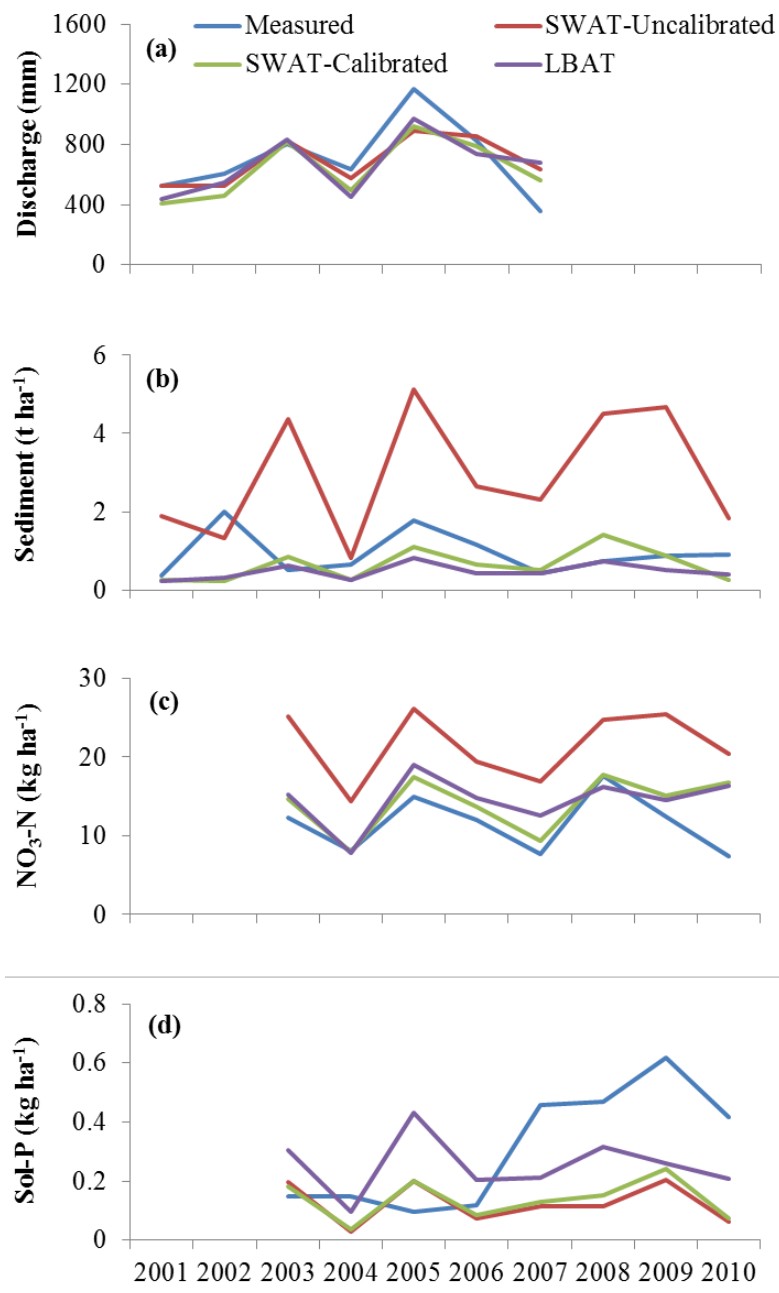


**Fig. 7** Simulations of annual stream discharge and sediment, $NO_3$-N, and Sol-P loadings

with LBAT and SWAT compared with field measurements at the outlet of LRW.




### 3.2.4  FDT Assessment in LRW


Mean annual water quantity and quality simulated with LBAT for agricultural lands of

LRW are shown in Table 13. The mean annual discharge for the baseline scenario was
626 mm greater than that for the six FDT scenarios (Table 13). When all agricultural
lands were protected (S6), there was a 2% reduction in discharge (equivalent to 11 mm;
Table 13). With the steepest areas protected (accounting for 10% of the total land base;
S1), the mean annual sediment loading was reduced by as much as 43% (equivalent to
4.5 t ha$^{-1}$; Table 13) and by as much as 81% (i.e., 8.57 t ha$^{-1}$) with all agricultural lands
protected (S6; Table 13). Mean annual Sol-P loading was reduced by 51% (equivalent to
0.47 kg ha$^{-1}$; Table 13). In contrast, increased usage of FDT tended to increase the mean
annual loading of $NO_3$-N, by about 6% when used across all agricultural lands
(equivalent to 1.73 kg ha$^{-1}$).





**Table 6.13** Impact of FDT on mean annual discharge and sediment, NO$_3$-N, and Sol-P
loadings simulated with LBAT under different FDT, provided in Table 3.

| Variable | Index | Baseline | S1 | S2 | S3 | S4 | S5 | S6 |
|---|---|---|---|---|---|---|---|---|
| Discharge (mm) | Mean | 626 | 625 | 623 | 622 | 619 | 616 | 615 |
| | MD | — | -1 | -2 | -4 | -7 | -10 | -11 |
| | PRD (%) | — | 0 | 0 | -1 | -1 | -2 | -2 |
| Sediment (t ha$^{-1}$) | Mean | 10.54 | 6.04 | 4.94 | 4.02 | 3.04 | 2.26 | 1.97 |
| | MD | — | -4.50 | -5.60 | -6.52 | -7.50 | -8.28 | -8.57 |
| | PRD (%) | — | -43 | -53 | -62 | -71 | -79 | -81 |
| NO$_3$-N (kg ha$^{-1}$) | Mean | 29.70 | 29.86 | 30.02 | 30.34 | 30.82 | 31.22 | 31.42 |
| | MD | — | 0.16 | 0.32 | 0.64 | 1.13 | 1.52 | 1.73 |
| | PRD (%) | — | 1 | 1 | 2 | 4 | 5 | 6 |
| Sol-P (kg ha$^{-1}$) | Mean | 0.94 | 0.89 | 0.83 | 0.76 | 0.65 | 0.52 | 0.46 |
| | MD | — | -0.05 | -0.11 | -0.17 | -0.28 | -0.42 | -0.47 |
| | PRD (%) | — | -5 | -11 | -19 | -30 | -45 | -51 |


Percentage change (based on PRD) of water quantity and quality were plotted against
percentage area of FDT for potato and barley in Fig. 8. Increasing the usage of FDT
helped to reduce discharge and sediment and Sol-P loadings for both crop types (Figs. 8a,
b, and c). It is worth noting that sediment loading decreased with increasing usage of
FDT (Fig. 16b). An opposite trend was observed for potato and barley with respect to the
impact of FDT on NO$_3$-N loading. With the increased usage of FDT, NO$_3$-N loadings
increased linearly for potato, while it decreased for barley. The increased for potato was
nearly twice as much as the reduction for barley (Fig. 16d). Seemingly the interaction
between barley and FDT had positive impacts on nitrate retention in soils, whereas the
interaction between potato and FDT had an opposite effect.
These results are consistent with the results from previous studies (Yang et al., 2012;
Yang et al., 2010), which used SWAT to assess the impact of FDT on water quantity and
quality within BBW. When using SWAT, greater efforts are needed to prepare basic


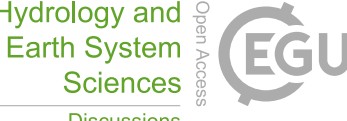


inputs, such as daily weather records, to proceed with its calibration and validation,
involving complex scenario setup and analysis. For every new watershed, SWAT needs
dedicated effort and time for its setup. LBAT, in contrast, can be used for multiple
watersheds as long as they have similar environmental conditions. Scenario analysis can
be directly conducted with different combinations of land use and BMPs using fewer
inputs than what is required by SWAT. Also, once developed, LBAT does not require
additional calibration.

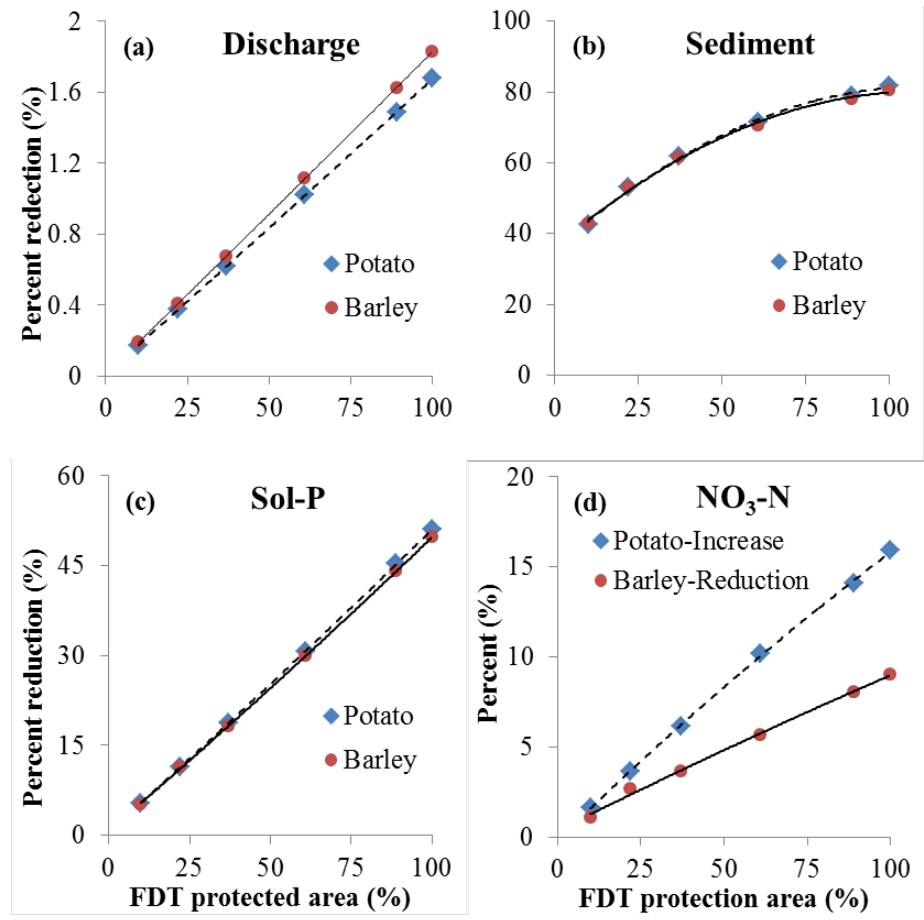


**Fig. 8** Percentage change in discharge and sediment, NO$_3$-N, and Sol-P loadings as a
function of % area, where FDT's were used.



## 4. Conclusion

The present study addresses the development of a decision support tool to assess the impact of land use change and BMPs on water quantity and quality for large ungauged watersheds. An enhanced version of SWAT was calibrated and validated for an experimental watershed. Multiple regression analyses were used to develop statistical equations based on simulations from SWAT. In total, three discharge and five sediment, $NO_3$-N, and Sol-P loading models were developed for different combinations of land use groups and BMP scenarios. Only four common predictors (i.e., annual precipitation, annual mean air temperature, mean saturated hydraulic conductivity of soil, and land use groups) and three unique predictors (LS-factor and annual nitrogen and phosphorus application rates for sediment, $NO_3$-N, and Sol-P loading models, respectively) are required.

With the aid of ArcGIS, statistical equations were integrated into the decision support tool, i.e., the land use and BMPs assessment tool (LBAT), whose basic simulation units are the DEM-grid cell. The LBAT was used to simulate annual water flow and sediment and nutrient loadings at the outlet of BBW. These simulations were compared with those of SWAT. LBAT and SWAT perform equally well. LBAT was subsequently applied to a large watershed (LRW). Results indicate that LBAT and the calibrated version of SWAT perform well with respect to annual stream discharge and sediment and $NO_3$-N loadings. LBAT performed slightly better, when Sol-P loading was considered. Compared with the uncalibrated version of SWAT, LBAT performed better. The impact of FDT on water quantity and quality was evaluated with LBAT for LRW; its results were consistent with the results generated with SWAT for the same region in previous studies. LBAT has



fewer input requirements than SWAT, and can be applied to multiple watersheds without
additional calibration. Also, scenario analyses can be directly conducted with LBAT
without complex setup procedures. We recommend using LBAT for economic analysis
and management decision making for watersheds with similar environmental conditions
of New Brunswick. The LBAT developed in this study may not be directly applied to
other regions; however, the approach in developing LBAT can be applied to other regions
of the world because of its flexible structure.

**Acknowledgement**
The funding support for this project was provided by Agriculture and Agri-Food Canada
(AAFC) through project #1145, entitled "Integrating selected BMPs to maximize
environmental and economic benefits at the field and watershed scales for sustainable
potato production in New Brunswick", and Natural Science and Engineering Research
Council (NSERC) through Discovery Grants to both CPAB and FRM. The research is
also partially supported by NASA (NNX17AE66G) and USDA (2017-67003-26485).
Authors are thankful to S. Lavoie, J. Monteith, and L. Stevens for their technical support
in data collection and sample analyses.

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
