# Peer review of "Developing a Decision Support Tool for Assessing Land Use Change and BMPs in Ungauged Watersheds Based on Decision Rules Provided by SWAT Simulation"

_Hydrology and Earth System Sciences, 2017_

## Referee Comment (RC1) · Anonymous Referee #1 · 20 Sep 2017

The manuscript entitled "Developing a Decision Support Tool for Assessing Land Use Change and BMPs in Large Ungauged Watersheds" presents development of decision support tool to estimate the impacts of land use change and best management practices on both water quantity and quality related issues of ungauged watersheds from Canada. The authors are putting their great efforts in this study. This type of research can help for making better informed decisions regarding future watershed management strategies. However, I have some comments and suggestions as given below, and want the authors to address thoroughly before considering this manuscript for further processes.

My major suggestions are: (1) Since calibration and validation of process-based models are crucial steps for further model simulation studies I suggest the authors to provide more details of these processes. (2) I expect to have some text about model parameters' sensitivity analysis and model prediction uncertainties. (3) I suggest to include more concrete outputs of the research in "Abstract" section, not the general statements. Specific suggestions are: ïČŸ Title: The term "Large Ungauged Watersheds" in the title is confusing to me because the larger watershed taken for this study is only 380 km2 and I don't find any statement to define a criteria whether a watershed is large or small in size. ïČŸ Abstract section, line 3: The term "water resources" should have some specifics. ïČŸ Materials and Methods section, line 104: "statistical equations". This should be clearly defined. ïČŸ Materials and Methods section, line 107: "water quantity and quality". These should be defined. ïČŸ Study Sites and Data Collection section, lines 123-124: I also want to include both minimum and maximum temperature and precipitation. ïČŸ Figure 3: I suggest making topographic slope in degrees. ïČŸ Study Sites and Data Collection section, lines 159-161: I suggest either to include website of data source or citation. ïČŸ Study Sites and Data Collection section, lines 161-162: I suggest to include more details. ïČŸ Modification of SWAT section, lines 176-177: Include some supportive document for this. ïČŸ SWAT Setup, Calibration, and Validation section, lines 197-198: Need more details of this. ïČŸ SWAT Setup, Calibration, and Validation section, lines 202-203: What are threshold values of land use, soil, and slope categories to define 32 sub-basins in the watershed? Need to explain. In summary, I find this manuscript within the scope of this journal and would like to go further if the authors would thoroughly address each suggestion provided above to better improve the manuscript from the present form.
* * *

---

## Referee Comment (RC2) · Anonymous Referee #2 · 24 Oct 2017

this study is a very interesting and important question for water resources management. However, I think this manuscript is not well prepared and is subjected to major revision for publication. I am reporting below some general comments and specific remarks, which I hope are useful. General comments: (1) The decision support tool should be established with readily available and measured variables only. Or, some advantages claimed in this study are not realistic. For instance, (a) anyone want to apply this method/framework to another catchment, they have to set up and calibrate the SWAT model first; (b) some of the explanatory variables might be catchment (sub-basin, or HRU) scale values and are un-observable, e.g. SOL_K, so regressed equation depends on the performance of the calibrated SWAT model. I suggest au-

thors to set up the tool independently with the SWAT model. Then, using the SWAT model to support the validity and to identify the advantages/disadvantages of the established tool. I think this is the way we usually do in operation, i.e. regressed and physically-based models are complementary and independent with each other for decision making. (2) I don't agree with the conclusion "DST and SWAT are equally well". The performance of DST and SWAT are "equally", which is not surprise as they are dependent, but not "well", which should be concluded on comparison with observations. Results did not well support "well". For the applications in the whole watershed, it is hard to say model was well established (or, it is just a numeric modelling experiment). (3) What is relationship of this study with four published studies of Qi et al. in term of modelling results of SWAT? If there is no new modification, set-up and calibration of the SWAT model, that is fine. But you have to say it explicitly and reduce the length of model introduction significantly. (4) Some general comments on the writing. Many abbreviations were used without full names where it was appeared firstly. Language should be edited carefully. Length should be reduced significantly (too many tables and figures). Suggest to separate the results and discussions. Subplots of all the figures should be labelled in order of (a), (b), . . . consistently. Specific comments: (1) Line 111: too many abbreviations in this flow chart. Consider move down to end of this section, or provide more specific information, or extend the caption. (2) Line 131: Provide information of all the abbreviations used in the figure in the captions. (3) Line 132: name of weather station should be consistent in form rather than one is "#08" and another one is "St. Leonard". (4) Line 139: The word "used by SWAT" is misleading. Land use and soil classes used by the SWAT model are much lesser (section 2.3) than these shown in this figure as many small patches of land cover and soil types are dissolved during the generation of HRUs. I suggest authors to provide the "real" and relevant information used by the SWAT (including information in table 3) rather than these maps/values based on raw datasets. (5) Line 148: what does "St. Quentin" mean? A type of soil? (6) Line 176-177: "It is believed that . . . even without calibration". How do I believe it? (7) Line 180: These two references are not the most relevant ones. (8) Line 193:

whether freeze-thaw cycles are considered here? Results said modelling error of sediment load was resulted from not considering freeze-thaw cycles in winter (line 507). (9) Line 193-194: what are "following changes"? How do I know the accuracy was improved? (10) Line 209: use four digital for the year consistently. (11) Line 313: delete "(LBAT)". (12) Line 350: what is (3)? (13) Line 484: In this section: it seems that results do not well support "increasing cell size increased sediment loading". Additionally, more explanations/discussions should be provided. (14) Line 486: Figure 13, where it is? (15) Line 508: "48" should be "48%". (16) Line 556: R2 should be included in this table.

---

## Author Comment (AC1) · 25 Oct 2017

Reviewer#1 The manuscript entitled "Developing a Decision Support Tool for Assessing Land Use Change and BMPs in Large Ungauged Watersheds" presents development of decision support tool to estimate the impacts of land use change and best management practices on both water quantity and quality related issues of ungauged watersheds from Canada. The authors are putting their great efforts in this study. This type of research can help for making better informed decisions regarding future watershed management strategies.

Thank you for your kind comment.

[Figure]

Since calibration and validation of process-based models are crucial steps for further model simulation studies I suggest the authors to provide more details of these processes. I expect to have some text about model parameters' sensitivity analysis and model prediction uncertainties.

We replied these comments along with several related topics in detail below.

I suggest to include more concrete outputs of the research in "Abstract" section, not the general statements.

We revised the abstract part according to your suggestion.

Title: The term "Large Ungauged Watersheds" in the title is confusing to me because the larger watershed taken for this study is only 380 km2 and I don't find any statement to define a criteria whether a watershed is large or small in size.

Compared with the small experimental watershed, the LRW is considered large. We accepted your suggestion and remove large from the tile to reduce confusion.

Abstract section, line 3: The term "water resources" should have some specifics

We revised that

Materials and Methods section, line 104: "statistical equations". This should be clearly defined.

We revised that.

Materials and Methods section, line 107: "water quantity and quality". These should be defined.

We revised that.

Study Sites and Data Collection section, lines 123-124: I also want to include both minimum and maximum temperature and precipitation.

We revised that.

Figure 3: I suggest making topographic slope in degrees.

We follow the setup of SWAT using percentage which is commonly used in SWAT papers.

Study Sites and Data Collection section, lines 159-161: I suggest either to include website of data source or citation.

We added website link.

Study Sites and Data Collection section, lines 161-162: I suggest to include more details.

We revised that.

Modification of SWAT section, lines 176-177: Include some supportive document for this.

We added references.

SWAT Setup, Calibration, and Validation section, lines 197-198: Need more details of this SWAT Setup, Calibration, and Validation section, lines 202-203: What are threshold values of land use, soil, and slope categories to define 32 sub-basins in the watershed? Need to explain.

We understand your suggestion on this part. However, we do not think adding more details regarding calibrated and validation SWAT for BBW and sensitivity analysis is necessary in the present paper as those processes can be find in a published paper (Qi et al. 2017b). Also the reviewer#2 has already pointed out that the paper needs to be shorten and more materials (which can be found easily in another paper) would not be helpful. The most important reason why we can not easily detail those processes in present paper is that the SWAT model was not just set up, calibrated and validated for BBW as did in other papers. We modified several modules in SWAT and tested them in separate papers and set up SWAT using filed-boundary based HRU configuration.

[Figure]

We think too much detail would divert readers attention from the objective of this paper.

Please also note the supplement to this comment:
https://www.hydrol-earth-syst-sci-discuss.net/hess-2017-423/hess-2017-423-AC1-supplement.pdf

―――――――――――――――

---

## Author Comment (AC2) · 25 Oct 2017

Reviewer# 2

This study is a very interesting and important question for water resources management. Thank you for your comments.

Major suggestions: (1) The decision support tool should be established with readily available and measured variables only. Or, some advantages claimed in this study are not realistic. For instance, (a) anyone want to apply this method/framework to another catchment, they have to set up and calibrate the SWAT model first; (b) some of the

explanatory variables might be catchment (sub-basin, or HRU) scale values and are un-observable, e.g. SOL_K, so regressed equation depends on the performance of the calibrated SWAT model. I suggest authors to set up the tool independently with the SWAT model. Then, using the SWAT model to support the validity and to identify the advantages/disadvantages of the established tool. I think this is the way we usually do in operation, i.e. regressed and physically-based models are complementary and independent with each other for decision making.

In general, we agree with your comments. We do want to develop a decision support tool based on measured variables only and then tested it by comparison with SWAT simulations. However, as we stated in the manuscript, it is almost impossible to get those measured data from field experiments (at least under the budget we have). Probably we could get a few regression equations from our limited field measurements, but they are insufficient to develop a watershed scale decision support tool which contains many land use and soil types and management practices and their combinations. To your specified questions: a) once a decision support tool was developed and validated under a specific climate, vegetation and soil conditions, the decision support tool could be used in many watersheds in that region. We do not need to setup and calibrate a SWAT model for each watershed we are interested in. This is one of advantages of DST over SWAT. For example, the decision support tool developed in the present study could be applied to many similar watersheds in New Brunswick. Without the DST, we probably have to setup SWAT model (or other watershed models) for each of them and then take long time to calibrate and validate models, which is not possible for ungauged watersheds (there are so many ungauged watersheds in New Brunswick) ; b) when we develop the decision support tool we chose physical meaningful variables. Sol_K is saturated hydraulic conductivity which is a standard measurement in many soil survey and maps. We do insist that SWAT simulation could provide information that are not available from field experiments. So, a well calibrated and validated SWAT model could provide more reliable information.

[Figure]

(2) I don't agree with the conclusion "DST and SWAT are equally well". The performance of DST and SWAT are "equally", which is not surprise as they are dependent, but not "well", which should be concluded on comparison with observations. Results did not well support "well". For the applications in the whole watershed, it is hard to say model was well established (or, it is just a numeric modelling experiment).

We agree with your comment. Both DST and SWAT were not performing very well compared with measurements. However, when it comes to ungauged watersheds, we do not even have measurements to validate the model. SWAT model has been used in many cases without calibration and decision makers still put some trust in its simulations because there is nothing else to consult to. The main purpose of present study it to provide a decision support tool for decision makers. At least, we could conclude that the DST performed equivalently as SWAT for the ungauged watershed and it is much easier to use than SWAT for decision makers.

(3) What is relationship of this study with four published studies of Qi et al. in term of modelling results of SWAT? If there is no new modification, set-up and calibration of the SWAT model, that is fine. But you have to say it explicitly and reduce the length of model introduction significantly.

To apply SWAT in Atlantic Canada region, modification of soil temperature, snowmelt and soil erosion modules are necessary to improve simulations of SWAT to develop DST for New Brunswick. We have revised this section to shorten the manuscript.

Many abbreviations were used without full names where it was appeared firstly. Language should be edited carefully.

We revised those issues as much as we can. Thanks

Length should be reduced significantly (too many tables and figures).

We put some results into appendix and delete several figures accordingly.

Suggest to separate the results and discussions

[Figure]

We understand your suggestion however we would like to keep results and discussion together to reduce manuscript length.

Subplots of all the figuresshould be labelled in order of (a), (b), : : : consistently

We revise them accordingly.

Specific suggestions: (1) Line 111: too many abbreviations in this flow chart. Consider move down to end of this section, or provide more specific information, or extend the caption

We removed the figure as it is confusing and not necessary in the manuscript. Thanks

2) Line 131: Provide information of all the abbreviations used in the figure in the captions

We revised them accordingly.

(2) Line 132: name of weather station should be consistent in form rather than one is "#08" and another one is "St. Leonard".

St. Leonard station is a national station while other stations are all local managed stations without a proper name. What they have is just a number.

(4) Line 139: The word "used by SWAT" is misleading. Land use and soil classes used by the SWAT model are much lesser (section 2.3) than these shown in this figure as many small patches of land cover and soil types are dissolved during the generation of HRUs.

We revised this part.

(5) I suggest authors to provide the "real" and relevant information used by the SWAT (including information in table 3) rather than these maps/values based on raw datasets.

The slope, soil and landuse maps are used to set up SWAT. Thanks

(5) Line 148: what does "St. Quentin" mean? A type of soil?

Yes, it is a type of soil.

(6) Line 176-177: "It is believed that : : : even without calibration". How do I believe it?
We revised it.

(7) Line 180: These two references are not the most relevant ones

We revised it.

(8) Line 193: whether freeze-thaw cycles are considered here? Results said modelling
error of sediment load was resulted from not considering freeze-thaw cycles in winter
(line 507).

Freeze-thaw cycles were considered by using modified version of SWAT in BBW and
LRW. However, the modified K-factor could not fully account for those processes. As
mentioned in Qi et al. 2017b, more studies are needed to address this issue in cold
regions.

(9) Line 193-194: what are "following changes"? How do I know the accuracy was
improved?

We revised the sentence. SWAT model Improvements could be referred to the four
papers of Qi et al.

(10) Line 209: use four digital for the year consistently.

We revised that.

(11) Line 313: delete"(LBAT)".

Yes.

(12) Line 350: what is (3)?

We revised it.

(13) Line 484: In this section: it seems that results do not well support "increasing cell

size increased sediment loading". Additionally, more explanations/discussions should be provided.

Those three sentences should be combined together to understand the fig 4. "Increasing cell size (i.e., slope length) increased sediment loading. However, the mean slope gradient was reduced. As a result, the mean sediment loadings were correlated nonlinearly with cell size as shown in fig 4".

(14) Line 486: Figure 13, where it is?

Typo. We revised it.

(15) Line 508: "48" should be "48%".

Yes.

(16) Line 556: R2 should be included in this table

We revised the table and added discussion about the results.

Please also note the supplement to this comment:
https://www.hydrol-earth-syst-sci-discuss.net/hess-2017-423/hess-2017-423-AC2-supplement.pdf

**Supplement:**

[revised manuscript text omitted]